Genome-wide identification and characterization of the KNOX gene family in Vitis amurensis

Liu Linling
Lu Wenpeng
Fan Shutian
Yang Yiming yangyiming@caas.cn
Institute of Special Animal and Plant Sciences, Chinese Academy of Agricultural Sciences , Changchun , China
Abd El-Moneim Diaa
Electronic publication date: 2025 Apr 9
Publication date: 2025
Volume: 13
Electronic Location ID: e19250
Received 2024 Oct 9; Accepted 2025 Mar 12
Copyright: ©2025 Liu et al.
Copyright year: 2025
Copyright holder: Liu et al.
License: This is an open access article distributed under the terms of the Creative Commons Attribution License, which permits unrestricted use, distribution, reproduction and adaptation in any medium and for any purpose provided that it is properly attributed. For attribution, the original author(s), title, publication source (PeerJ) and either DOI or URL of the article must be cited.
License URL: https://creativecommons.org/licenses/by/4.0/

Keywords: KNOX gene family, Vitis amurensis, Expression analysis, Genome-wide identifications, Bioinformatic

Funding: The Agricultural Science and Technology Innovation Program CAAS-ASTIP-2021-ISAPS This work was supported by the Agricultural Science and Technology Innovation Program (CAAS-ASTIP-2021-ISAPS). The funders had no role in study design, data collection and analysis, decision to publish, or preparation of the manuscript.

==============================
Background

The KNOX (KNOTTED1-like homeobox gene) gene family plays a pivotal role in controlling plant growth, maturation, and morphogenesis. However, the function of KNOX in Vitis amurensis has not yet been reported. This study identified and characterized the entire KNOX gene family in Vitis amurensis.

Methods

By employing bioinformatic approaches, the phylogenetic relationships, chromosomal positions, gene architectures, conserved motifs, cis-regulatory elements present in promoter regions, and gene expression profiles of KNOX gene family members in Vitis amurensis were identified and analyzed.

Results

Ten KNOX genes spanning nine chromosomes were discovered, and these genes were subsequently categorized into two distinct subclasses. The promoter regions of members of the KNOX gene family include cis-acting elements that are involved in plant growth, hormonal regulation, and stress and light responses. An examination of the expression profiles of KNOX genes in different tissues of Vitis amurensis revealed that genes in Class I presented tissue-specific expression patterns, whereas genes belonging to Class II presented more ubiquitous expression across various tissues. The expression levels of Vitis amurensis KNOTTED1-like homeobox (VaKNOX)2, VaKNOX3, and VaKNOX5 were highest in fruits. VaKNOX2, VaKNOX3, and VaKNOX5 can serve as candidate genes for enhancing fruit quality. The expression levels of VaKNOX6 and VaKNOX7 were much higher in cold environments than in normal conditions. Through in-depth research into the functions of VaKNOX6 and VaKNOX7, we aimed to improve the cold resistance of grapevine varieties.

Introduction

The KNOX gene family, alternatively referred to as KNOTTED1-like homeobox genes, encodes transcriptional regulators that consist of homeobox proteins. These proteins are abundant in plants and play pivotal roles in controlling plant growth, maturation, and morphogenesis (Hay & Tsiantis, 2009). Corn Knotted-1 (Kn1), the first homeobox gene discovered, is highly important in plant molecular biology (Vollbrecht et al., 1991). KNOX genes have been discovered in a growing number of plants and exhibit high sequence similarity and similar functions to those of corn Knotted-1. The widespread occurrence of the KNOX gene family across plants indicates its vital role in plant evolution. The proteins encoded by KNOX genes have four domains: KNOX1, KNOX2, ELK, and Homeobox KN (Hay & Tsiantis, 2010a; Hay & Tsiantis, 2010b). Based on the structural attributes and evolutionary trends of KNOX genes, the KNOX gene family in dicotyledonous plants can be grouped into three distinct categories: Class I, Class II, and Class KNATM (Reiser, Sánchez-Baracaldo & Hake, 2000; Barton, 2001). Class I subfamily genes represent an essential category of genes that form and maintain and form plant meristems, are expressed mainly in the apical meristems of plants, and affect the morphogenesis of tissues and organs by regulating cell differentiation related to plant organ formation (Tsuda & Hake, 2015). In Arabidopsis, the Class I subfamily member SHOOT MERISTEMLESS (STM) gene controls the differentiation of the entire apical meristem and promotes the proliferation of stem cells to form new leaves (Abraham-Juárez et al., 2010). In tomatoes, the Class I subfamily members Tkn1 and Tkn2 are expressed during the initial formation of young bud primordia (Jasinski et al., 2007). The overexpression of Class I genes in tomato leaves increases the complexity of the leaves and promotes defects in leaf margins and other morphological aberrations. Plants with mutations in both the osh1 and osh15 genes in Class I rice allow for only the formation of leaf-like structures in the callus, and no buds can form (Chen et al., 2023; Niu et al., 2022). Class II subgroup genes exhibit relatively pervasive expression patterns across diverse tissues and organs, and primarily functioning in the regulation of plant organ differentiation and the biosynthesis of secondary cell walls. KNOX4 genes belonging to Class II in Arabidopsis may play a role in seed dormancy in mutant varieties (Iannelli et al., 2023). Both KNAT3 and KNAT7 play roles in regulating mucilage biosynthesis in Arabidopsis seeds. Mucilage is a sticky polysaccharide-rich substance that coats the seed surface and plays a crucial role in seed dispersal and germination (Zhang et al., 2022a; Zhang et al., 2022b). By regulating the expression of genes involved in mucilage production and excretion, KNAT3 and KNAT7 contribute to the development and properties of this crucial seed constituent. Class II genes orchestrate various molecular processes to influence the growth and development of fruits as well as fruit traits (Shtern et al., 2023). In tomatoes, Class II genes also affect the ripening of pulp (Keren-Keiserman et al., 2022). The KNATM class is a unique subclass of KNOX genes in dicotyledonous plants (Gao et al., 2015). In Arabidopsis, KNATM genes are involved in regulating leaf polarity and leaf traits (Luo et al., 2012). KNATM is uniquely expressed in the proximal–lateral areas of organ primordia as well as at the edges of fully developed organs (Magnani & Hake, 2008). Consistent with this finding, genetic studies have elucidated the role of KNATM in shaping the proximal−distal patterning of leaves. Detailed in vivo domain analyses have emphasized the functional domains of KNATM. The role of KNATM as a transcriptional regulator has been well established through in vivo domain analyses. The functions of KNOX genes in plants are diverse. They regulate cell differentiation, thereby influencing the formation of plant tissues and the development of organs (Bueno et al., 2021). Additionally, KNOX genes participate in plant hormone signal transduction and stress responses and interact with other transcription factors, thus collectively regulating the growth and development of plants (Scofield & Murray, 2006). Overexpression of the TaKNOX14-D gene produced high resistance to cold stress (Li et al., 2022). Given the critical roles of KNOX genes in plant development and their evolutionary conservation, investigating the KNOX gene family in Vitis amurensis could provide insights into the molecular mechanisms driving its unique traits, particularly those relevant to fruit production and stress resistance.

Vitis amurensis belongs to the genus Vitis within the family Vitaceae and is primarily concentrated in northeastern China, with significant distributions in Jilin, Liaoning, and Inner Mongolia (Wang et al., 2021a; Wang et al., 2021b; Ma et al., 2022). This vine plant, characterized by long vines and reddish-brown bark, is notable for its resveratrol content, which reduces the risk of hyperlipidemia, coronary heart disease, and thrombosis, and has anticancer properties (Zhao, Duan & Wang, 2010). V. amurensis, an excellent cold-resistant germplasm resource, is characterized by enduring temperatures as low as −40 °C, making it a crucial material for exploring key cold resistance genes in grapes and elucidating the molecular mechanisms of frost tolerance. There is limited knowledge regarding the composition of the KNOX gene family in V. amurensis. Despite its importance, the composition and functional roles of the KNOX gene family in V. amurensis remain poorly understood. To address this knowledge gap, we conducted a comprehensive analysis of the KNOX gene family members in V. amurensis, focusing on their identification, sequence characteristics, evolutionary relationships, and expression patterns. Our study aimed to provide deeper insights into the diverse functions of KNOX genes, particularly their involvement in cold tolerance mechanisms. By analyzing the roles of KNOX gene family members in cold resistance, this research establishes a theoretical foundation for the molecular breeding of cold-resistant grape varieties. These findings not only enhance our understanding of the genetic basis of cold resistance in V. amurensis but also contribute to the broader field of grape genetics and breeding research, offering valuable resources for developing resilient grape cultivars in the face of climate challenges.

Materials and Methods

Identification of KNOX genes

Genomic and proteomic sequence information for V. amurensis was downloaded from the TCMPG 2.0 database under number TCMPG20325 (Wang et al., 2024).

Hidden Markov models (HMMs) for the KNOX gene family were obtained from the Pfam database (Mistry et al., 2021). Two models, KNOX1 (PF03790.16) and KNOX2 (PF03791.16), were utilized. HMMER software was used to search for KNOX protein sequences within the proteome of V. amurensis. The E value was less than 1e−5 in the HMMER search. Ten sequences were found in the HMMER search. Sequence files for the KNOX proteins were retrieved from the TAIR database (Poole, 2007). BLAST software was used to align these sequences with the V. amurensis database (Altschul et al., 1990). The E value was less than 1e−5 in the BLAST search. Twenty sequences were found in the BLAST search. The resulting sequences were merged, and duplicates were removed to identify the KNOX gene family in V. amurensis.

Bioinformatics analysis

The physicochemical attributes of members of the KNOX gene family in V. amurensis were investigated using the online ExPASy tool (Duvaud et al., 2021). The subcellular localization of the KNOX gene family members was determined using the software tool CELLO. The chromosomal localization of the KNOX genes was accurately determined using TBtools v 2.119 software (Chen et al., 2020).

The protein domains of the KNOX genes in V. amurensis were examined using the CDD website and the Pfam database. Default search parameters were employed for domain identification. Multiple sequence alignments of KNOX protein sequences were performed using MUSCLE in MEGA11, and the alignments were displayed with Jalview software.

MEGA11 software was used for sequence alignment (Tamura, Stecher & Kumar, 2021), which was followed by the construction of a phylogenetic tree via the neighbor-joining (NJ) method with a bootstrap value of 1,000 and default settings for all remaining parameters. The phylogenetic tree was subsequently refined and visualized using the online tool iTOL.

To further study the evolutionary relationships of the KNOX genes among species, synteny analysis was conducted on V. amurensis and A. thaliana with TBtools v 2.119 software. The gaps were completely deleted.

Using the results of the genomic annotation of V. amurensis, a comprehensive gene structure analysis of KNOX family members was carried out via TBtools v 2.119 software.

Using TBtools v 2.119 software, the upstream 2000 bp promoter regions of the CDSs for each KNOX gene were extracted from the V. amurensis genomic database. The cis-acting regulatory elements within the promoters of the KNOX genes were subjected to rigorous analysis. Therefore, the sequences of the promoter regions were uploaded to the PlantCARE database for the recognition and enumeration of diverse cis-acting elements (Lescot et al., 2002).

Expression patterns of the VaKNOX genes in various tissues of V. amurensis

Vitis amurensis ‘Zuoshan 1’ is a variety of Chinese wild grape cultivated by the Institute of Special Animal and Plant Sciences of CAAS. Six-year-old Vitis amurensis Rupr ‘Zuoshan 1’ plants were planted in the National Germplasm Repository. The National Germplasm Repository is located in the town Zuojia, Jilin city, Jilin Province (43.57°N, 125.59°E), which possesses a temperate continental monsoon climate. The growth conditions of Vitis amurensis ‘Zuoshan 1’ are adapted to the local natural conditions. The buds, old leaves, tendrils, seeds, roots, fruits, flowers, and stems of Vitis amurensis ‘Zuoshan 1’ were collected. Immediately after collection, the samples were chilled in liquid nitrogen and subsequently preserved in a −80 °C freezer. All of the samples were tested in triplicate, and the experiments were performed on three biological replicates. The stems were used as the control group for comparative purposes.

Expression patterns of VaKNOX genes under cold conditions

Stem samples were collected in summer, when the outdoor temperature was 20 °C. Stem samples were collected in winter, when the outdoor temperature was −20 °C. Immediately after collection, the samples were chilled in liquid nitrogen and subsequently preserved in a −80 °C freezer. All samples were tested in triplicate, and the experiments were performed on three biological replicates. The summer conditions were considered normal and were used as the control group for comparative purposes. The winter conditions were considered to represent cold conditions.

RNA extraction and reverse transcription qPCR (RT–qPCR)

Total RNA was extracted from the samples using an RNA extraction kit (Nanjing Vazyme Biotech) according to the manufacturer’s instructions. The RNA extraction kit contains buffer RWA and buffer RWB, which can remove DNA, phenol, or polysaccharides. The concentration and purity of the extracted RNA were assessed using a Biochrom Bio Drop Duo, and its integrity was detected  via 1% agarose gel electrophoresis. The RNA concentration for each sample with an A260/280 value ranged from 1.89 to 1.99, indicating that the extracted RNA was of high quality. Finally, 800 ng of RNA was efficiently reverse transcribed into cDNA using a specialized kit (Nanjing Vazyme Biotech).

Based on the basis of the coding sequences (CDs) of the KNOX genes in V. amurensis, specific primers for fluorescent quantitative amplification were designed using Primer 5.0 software, with the actin gene (Gene ID: mRNA: Vitis04G1372) as a control (Table 1). The primers used were synthesized by Sangon Biotech (Shanghai). The expression levels of the genes were measured with the Real-time PCR Super Mix SYBR Green Kit (Beijing TransGen Biotech). The reaction mixture consisted of 20 µL of 2 ×Taq PCR Master Mix (10 µL), ddH2O (eight µL), upstream and downstream primers (each at 10 µmol/L, 0.5 µL each), and template cDNA (one µL). The PCR program was as follows: initial denaturation at 95 °C for 5 min, followed by 40 cycles of denaturation at 95 °C for 10 s, annealing at 60 °C for 10 s, and fluorescence collection at 60 °C for 10 s. All the experiments were performed in three biological replicates, and each replicate was measured three times. In all the experiments, the specifications of the no-template controls (NTCs) and the no-reverse transcription controls (NRTs) were determined. Relative expression levels were analyzed via the 2−ΔΔCt method (Livak & Schmittgen, 2001). The raw data were analyzed using Microsoft Excel software, and GraphPad Prism 9.5 was used for data analysis and graphical visualization. We analyzed RT−qPCR data for 10 KNOX genes in various tissues of V. amurensis using an analysis of variance (ANOVA). We analyzed the RT−qPCR data of 10 KNOX genes under normal and cold conditions using t tests.

Results

Identification and chromosome localization of V. amurensis KNOX genes

Using the BLAST and HMMER methods, we conducted a thorough search of the V. amurensis database. We subsequently verified the identified sequences to confirm their accuracy and relevance to the KNOX gene family. This comprehensive analysis revealed the presence of ten KNOX gene family members in V. amurensis. The amino acid counts of the identified KNOX proteins varied significantly, ranging from 175–500. The molecular weights of these proteins also varied widely, ranging from 19.37 to 55.18 kDa. Additionally, the isoelectric points of the proteins varied between 4.64 and 6.23. Subcellular localization predictions via reliable prediction tools unanimously revealed that all 10 KNOX genes were localized in the nucleus (Table 2).

Table 1 Primer sequences.

Gene name	Forward primer sequence (5′–3′)	Reverse primer sequence (5′–3′)	Product size	
Actin	CTTGCATCCCTCAGCACCTT	TCCTGTGGACAATGGATGGA	162	
VaKNOX1	GGCTCTTACGGAGGTTTGGTGGTC	TTTCCTGTGCCGCTTTCGTTGAT	183	
VaKNOX2	TTTCCTGTGCCGCTTTCGTTGAT	AGAGTGAGTTGCTGTGTTGGTGCTG	211	
VaKNOX3	CAGATTTGGTCCTCTTGTCCCTA	CAACTTTGCCTTGTCCTACTCAG	196	
VaKNOX4	CCATACCCATCGGATTCACAGA	GGAAAGGGATTGCCCAAGACAT	184	
VaKNOX5	GCGATAACAATGGTGAGGACAAG	CTTGGTTACAACGCCCTGAGAAT	236	
VaKNOX6	GAGGAGGAGGTTGATGGGAATGA	CCGATGGGTAAAGCCACTTGTAG	249	
VaKNOX7	TGATTTCTCCGTCTCACTACCCG	TCTTCTGCAAAGCTCGCTTCGTT	182	
VaKNOX8	GGAAAGGGATTGCCCAAGACAT	AAGTCGCTTCATCTAAGGGTTTC	235	
VaKNOX9	AAGTCGCTTCATCTAAGGGTTTC	ACCACCAAAGACGACGACGACAA	255	
VaKNOX10	TCATCTTCATTCGGGCGAGTGTT	TTCGTTCCCTTGTTGTTTTTGCT	217	

Table 2 Basic information and subcellular localization of the VaKNOX genes.

Gene name	Gene ID	Chromosome location	Number of amino acids	Relative molecular weight (103)	pI	Signal peptide	Subcellular localization	
VaKNOX1	Vitis01G0695	1	318	36.29	4.81	no	Nucleus	
VaKNOX2	Vitis02G0413	2	293	33.29	6.23	no	Nucleus	
VaKNOX3	Vitis04G0555	4	448	50.13	5.24	no	Nucleus	
VaKNOX4	Vitis10G0018	10	361	40.45	6.01	no	Nucleus	
VaKNOX5	Vitis11G0624	11	500	55.18	5.51	no	Nucleus	
VaKNOX6	Vitis12G0428	12	359	39.98	6.10	no	Nucleus	
VaKNOX7	Vitis12G0772	12	206	22.47	4.98	no	Nucleus	
VaKNOX8	Vitis14G0107	14	175	19.37	4.64	no	Nucleus	
VaKNOX9	Vitis17G0805	17	319	35.87	5.14	no	Nucleus	
VaKNOX10	Vitis18G0602	18	370	42.52	5.80	no	Nucleus	

The chromosome localization results revealed that the 10 KNOX proteins in V. amurensis were dispersed across nine distinct chromosomes (Fig. 1). Specifically, VaKNOX1 is located on chromosome 1, VaKNOX2 is located on chromosome 2, VaKNOX3 is located on chromosome 4, VaKNOX4 is located on chromosome 10, VaKNOX5 is located on chromosome 11, VaKNOX6 and VaKNOX7 are located on chromosome 12, VaKNOX8 is located on chromosome 14, VaKNOX9 is located on chromosome 17, and VaKNOX10 is located on chromosome 18.

Multiple sequence alignment

The alignment of multiple sequences revealed that the VaKNOX protein sequences possessed four conserved domains, which were specifically designated KNOX I, KNOX II, ELK, and Homeobox KN (Fig. 2). Among them, the Homeobox KN domain exhibited the highest conservation, showing the characteristic structure of the TALE homeobox protein superfamily with three extra amino acids (P-Y-P). The domains of KNOX I, KNOX II, and ELK also demonstrated good conservation. KNOX II has an E-L-D sequence, and the conserved ELK domain has an E-L-K sequence (Fig. 3).

Figure 1 Chromosome mapping of VaKNOX genes.

Figure 2 Alignment of multiple sequence of VaKNOX proteins.

Figure 3 HMM logos of conserved domains in VaKNOX proteins.

(A) KNOXII, (B) KNOXI, (C) ELK, (D) Homeobox KN.

Systematic evolutionary analysis and collinearity analysis

Using the KNOX protein sequences from A. thaliana and V. amurensis, we constructed an evolutionary tree (Fig. 4). Within this tree, the KNOX proteins were categorized into three distinct subfamilies: Class I, Class II, and Class KNATM. Class I included four A. thaliana KNOX proteins and seven V. amurensis KNOX proteins, whereas Class II included four A. thaliana KNOX proteins and three V. amurensis KNOX proteins.

Figure 4 Evolutionary tree of the KNOX protein sequences.

Intermediate synteny analysis of the KNOX gene family in V. amurensis and A. thaliana revealed 10 pairs of syntenic genes (Fig. 5). VaKNOX4, VaKNOX7 and VaKNOX8 have no syntenic relationship. All other VaKNOX genes presented syntenic relationships.

Figure 5 Collinearity analysis between KNOX in Vitis amurensis and Arabidopsis thaliana.

Gene structure and conserved protein domains

Using the intraspecific phylogenetic tree as a foundation, we examined the gene structure and dispersal of conserved protein domains within the KNOX genes of V. amurensis (Fig. 6). Our findings revealed that the VaKNOX family genes are primarily composed of two to six exons and one to four introns. The length of introns and the number of exons varied among different classes. Most VaKNOX proteins contained the conserved protein domains KNOXI, KNOXII, ELK, and Homeobox KN, which are arranged in sequence. A few genes lacked one to two conserved domains, with VaKNOX2 lacking the ELK domain and VaKNOX7 and VaKNOX8 lacking both the ELK and Homeobox KN domains.

Cis-acting elements in the VaKNOX proteins

To better understand the mechanisms underlying the regulation of VaKNOX protein expression in V. amurensis, we predicted the cis-acting elements associated with each gene family member.

The results indicated that the promoter regions of each member contained a considerable quantity of cis-acting elements, with significant variations in both quantity and type. A comprehensive prediction of the 10 VaKNOX genes revealed a total of 33 types of cis-acting elements, which are involved primarily in plant hormones, growth, stress responses and light responses (Fig. 7).

Figure 6 Gene structure and conserved protein domains of VaKNOX proteins.

Figure 7 Cis-acting elements in the VaKNOX proteins of V. amurensis.

(A) Differently coloured histograms represent the sum of the four types of cis-acting elements in the VaKNOX gene; (B) heatmap of the cis-acting elements in each VaKNOX gene, with different colours indicating the quantity of cis-acting elements and white representing the absence of cis-acting elements.

All 10 VaKNOX proteins were found to be involved in stress and light responses. There were 13 distinct types of light-responsive elements, such as Box 4, the G-box, and Sp1. There were seven types of stress-responsive elements, including abscisic acid responsive elements (ABREs), anaerobic response elements (AREs), and adenine and thymine (AT)-rich elements. There were seven types of elements related to plant growth and development, including A-boxes, O2-sites, and RY-elements. There were six types of hormone-responsive elements, including the CGTCA motif. These results suggest that VaKNOX genes may play a significant role in plant hormones, growth, stress responses and light responses and that there may be time- and tissue-specific effects on VaKNOX gene expression.

Expression patterns of the VaKNOX genes in various tissues of V. amurensis

Using RT−qPCR, we analyzed the expression patterns of 10 KNOX genes in various tissues of V. amurensis, including buds, old leaves, tendrils, seeds, roots, fruits, flowers, and stems (Fig. 8). An examination of the expression profiles of KNOX genes in different tissues of Vitis amurensis revealed that genes in Class I presented tissue-specific expression patterns, whereas genes belonging to Class II presented more ubiquitous expression across various tissues.

Figure 8 Expression patterns of the VaKNOX genes.

Different lowercase letters indicate significant differences in the expression of the same gene in same tissue at different time points (P < 0.05). Different capital letters indicate significant differences in the expression of the same gene in same tissue at different time points (P < 0.01).

The Class I genes were highly expressed in buds and tendrils. Extremely high expression levels of VaKNOX4 and VaKNOX6 were detected in the seeds. The expression levels of VaKNOX2, VaKNOX3, and VaKNOX5 were highest in fruits.

Expression patterns of the VaKNOX genes under cold conditions

Using RT−qPCR, we determined that VaKNOX genes presented different expression patterns under normal and cold conditions (Fig. 9). There was no difference in the expression of the VaKNOX3, VaKNOX8, or VaKNOX10 genes between normal and cold conditions. The expression levels of VaKNOX1, VaKNOX2, VaKNOX4, VaKNOX5, and VaKNOX9 were much greater under normal conditions than under cold conditions. Conversely, the expression levels of VaKNOX6 and VaKNOX7 were much higher under cold conditions than under normal conditions.

Figure 9 Expression patterns of the VaKNOX genes in cold condition.

Asterisks *,** and *** indicate significant differences compared with the EV at p < 0. 05 p < 0.01 and p < 0.001 (Student’s t-test), respectively.

Discussion

The KNOX gene family is an integral group of transcription factors (Sun et al., 2023). These genes not only regulate cell differentiation and help maintain the activity of meristematic tissues, but also regulate key biological processes such as organ morphogenesis (Hake et al., 2004). Despite extensive research on the KNOX gene family in various plants, reports on the KNOX genes of V. amurensis remain limited.

In this comprehensive genomic study, we investigated the KNOX gene family of V. amurensis, which included a total of 10 distinct members. This number surpasses that of A. thaliana, which has nine members, and that of Solanum lycopersicum, which has eight members, but is less than the 13 members in Oryza sativa and the impressive 27 impressive members in Glycine max (Wang et al., 2021a; Wang et al., 2021b). The variation in the number of KNOX genes in Vitis amurensis reflects the balance of multiple evolutionary pressures that it has experienced during its evolution and is correlated with its unique growth habits. All the identified KNOX genes were predicted to be localized in the nucleus, which is consistent with the function of KNOX proteins as transcription factors. The nucleus is a crucial site for gene expression regulation, where KNOX proteins participate in regulating the expression of target genes by binding to DNA or interacting with other transcription factors. In many instances, KNOX proteins interact with each other. Arabidopsis KNAT3 can form homodimers with itself or heterodimers with KNAT7, facilitating the regulation of various developmental stages (Qin et al., 2020). Apple MdBLH2 interacts with the Class I KNOX protein MdKNOX15, resulting in the reduced binding and ability of MdKNOX15 to activate the downstream target gene MdGA2ox7 promoter (Jia et al., 2021). The distribution of the KNOX gene family members across nine different chromosomes revealed the complexity and diversity of this family in the Vitis amurensis genome. The chromosome localization results also provide clues for studying the evolutionary history of the KNOX gene family, which can reveal the roles of gene duplication, recombination, and loss in the evolution of the family. This variation in gene count across species underscores the rich diversity and complexity of the KNOX gene family across the plant kingdom. By performing a meticulous analysis of protein sequences, structural features, and phylogenetic relationships, we categorized these VaKNOX genes into two distinct classes: Class I and Class II (Frangedakis et al., 2017). Intriguingly, VaKNOX1, VaKNOX7, VaKNOX8, VaKNOX9, VaKNOX10, VaKNOX6, and VaKNOX4 clustered tightly with the Class I genes of A. thaliana, indicating a shared evolutionary lineage and potentially conserved functional roles. Moreover, the remaining VaKNOX genes were grouped with the Class II genes of Arabidopsis, suggesting unique functional traits within this subclass. Of particular interest were VaKNOX6 and VaKNOX4, which belong to Class I and exhibited a marked phylogenetic proximity to AtSTM-AT1G62360 from A. thaliana, involved in shoot apical meristem (SAM) formation and maintenance of the stem cell population, floral and carpel formation (Endrizzi et al., 1996). VaKNOX6 and VaKNOX4 may have similar biological functions in the establishment of the shoot apical meristem.

Within the intricate KNOX gene family of V. amurensis, an array of conserved domain variations has been observed among its diverse members. Typically, most of these genes exhibit four core protein domains: KNOX I, KNOX II, ELK, and Homeobox KN. However, VaKNOX2, which lacks the ELK domain, stands out as an exception, whereas VaKNOX7 and VaKNOX8 are distinguished by the absence of both the ELK and Homeobox KN domains. The ELK domain, which serves as a nuclear localization signal (NLS), spans approximately 21 amino acids and is rich in glutamic acid (Glu, E), leucine (Leu, L), and lysine (Lys, K) (Hofer et al., 2001). This ELK domain can function as a nuclear localization signal involved in transcriptional repression, whereas the homeodomain may play a role in recognizing promoter sequences of downstream genes (Bueno, Alvarez & Ordás, 2020). These variations in structural domains are likely to confer unique biological roles and functions to each gene, adding to the complexity and specialization within this gene family.

Cis-acting elements, which are specific DNA sequences that are arranged in tandem with structural genes, are integral components of the transcriptional regulatory machinery (Jia et al., 2023). All ten VaKNOX genes were found to contain elements associated with photoresponsiveness and stress responsiveness. These findings suggest that these genes are likely involved in photomorphogenesis, the process of plant development in response to light, and the adaptive modulation of biotic or abiotic stressors in V. amurensis. A plethora of cis-acting elements associated with light responsiveness were detected in the promoter region of the orchid KNOX gene and were preceded by methyl jasmonate (MeJA)- and abscisic acid (ABA)-responsive elements. This discovery suggested that the KNOX gene plays a pivotal role in regulating both light response mechanisms and hormonal signaling in plants (Zhang et al., 2022b). Additionally, the promoter sequence located 2000 bp upstream of the coding region of the apricot KNOX gene may participate in crucial physiological processes, including metabolism, growth, and development, within plants (Bai et al., 2023). A thorough analysis of the promoter elements in tea plants revealed that most KNOX genes respond to diverse stimuli, including drought, salinity, cold temperatures, and exogenous applications of methyl jasmonate and gibberellin (Dai et al., 2023). The VaKNOX gene family contains six types of hormone-responsive elements, suggesting that the expression levels of these genes fluctuate in response to hormonal influences (Hay & Tsiantis, 2010a; Hay & Tsiantis, 2010b). Existing cis-acting elements can either alter the trans-regulatory landscape or reconfigure KNOX expression through modifications in cis-regulatory elements (Li et al., 2023). The interplay between different cis-acting elements, such as light-responsive and hormone-responsive motifs, could create complex regulatory networks that dynamically control gene expression by integrating multiple signaling pathways. Future research should focus on functional validation of these elements through targeted mutagenesis and transcriptional assays, as well as comparative analyses across diverse V. amurensis species, to uncover conserved regulatory motifs and species-specific adaptations, ultimately informing strategies to increase stress tolerance and resilience in V. amurensis through molecular breeding.

A meticulous analysis of the expression patterns of the VaKNOX gene family across diverse tissues of V. amurensis revealed intricate tissue-specific expression profiles among its members. Notably, genes belonging to Class I of the VaKNOX family presented preferential expression in buds and tendrils. Studies have shown that the KNOX I gene family functions in maintaining meristems and functions primarily in the development of lateral organs (Stammler et al., 2013). Single-gene mutants exhibit significant defects in leaf development and prominent morphological changes.Research in Chrysanthemum × morifolium has shown that modulating the activity of KNOX genes can regulate the production and distribution of axillary buds (Yang et al., 2023). The stm mutant in Arabidopsis thaliana and the kn1 mutant in Zea mays exhibit similar defects in shoot apical meristem development, both of which fail to form stem apical meristems with normal differentiation capabilities (Bhatt et al., 2004). A gain of KNOX function in species where KNOX genes are part of a compound leaf development program can lead to a striking reiteration of leaflets (Uchida et al., 2010). Analysis of the gene regulatory networks involving STM, KN1, and their rice ortholog ORYZA SATIVA HOMEOBOX1 (OSH1) proteins indicates that Class I KNOX proteins target not only components of the gibberellic acid (GA) and cytokinin (CK) pathways but also those of other hormones such as auxin and brassinosteroids (BRs) (Scofield et al., 2018). Class I KNOX factors preserve stem cell pluripotency by regulating hormone activity networks to ensure low levels of GA and high levels of CK within the meristem (Jasinski et al., 2005). Sustained stem cell activity enables plants to optimize their leaves. Therefore, Class I KNOX genes have specific functions in compound leaf development that are distinct from their ability to induce shoot meristem formation. VaKNOX4 and VaKNOX6 are highly expressed during seed development, indicating their potential involvement in seed growth and development. VaKNOX2, VaKNOX3 and VaKNOX5, which belong to Class II, exhibited high expression levels across various tissues, which is consistent with previous observations that Class II subfamily genes have broad expression patterns. Class II KNOX genes display diverse expression patterns, and their precise functions have mostly remained unknown until recently. The high expression levels of VaKNOX2, VaKNOX3 and VaKNOX5 in fruits may indicate that these genes play a role in fruit development and maturation. These findings are consistent with those of previous studies in Arabidopsis thaliana and Solanum lycopersicum, which identified Class II KNOX genes as regulators of fruit size, shape, and maturation (Furumizu et al., 2015; Keren-Keiserman et al., 2022). Recent examination of the roles of SlCLASSII KNOX (TKN-II) genes roles in tomato fruit development has shown that the tknII3, tknII5, and tknII7 single mutants significantly reduce fruit size and the shape index. The reduced number of pericarp cell layers and altered fruit shapes in 35S:amiRTKN-II fruits could be bypassed by the procera mutation, which triggers a constant response to the plant hormone GA. Neither the procera mutation nor the application of ectopic GA fully restored the pericarp cell size characteristics of the TKN-II mRNA-knockdown fruits. These discoveries imply that TKN-II genes regulate fruit through both GA-dependent and GA-independent mechanisms (Shtern et al., 2023). In summary, analysis of the VaKNOX gene family in Vitis amurensis vis tissue-specific expression provides valuable information on the potential functional roles of these genes. Owing to their preferential expression in buds and tendrils, Class I VaKNOX genes appear to play a critical roles in maintaining meristems and regulating lateral organ development, which is consistent with the well-documented functions of KNOX I genes in other species. The high expression levels of VaKNOX4 and VaKNOX6 during seed development suggest their involvement in seed growth and germination, suggesting potential opportunities for improving seed vigor under suboptimal conditions. Moreover, the broad tissue-specific expression of Class II VaKNOX genes, particularly in fruits, hints at their broader regulatory roles in fruit development and maturation. However, the lack of functional data for these genes in Vitis amurensis underscores the need for targeted mutagenesis and reverse genetics approaches to validate their proposed roles.

Vitis amurensis, known as the mountain grape, has the strongest cold tolerance among grape species and serves as a crucial source of cold resistance genes. When the wheat gene TaKNOX14-D was transferred, Arabidopsis presented increased resistance to cold stress. Under cold stress conditions, certain wheat KNOX genes are upregulated, indicating their potential involvement in the plant cold stress response (Li et al., 2022). VaKNOX6 and VaKNOX7 were highly expressed under cold stress conditions. The high expression levels of VaKNOX6 and VaKNOX7 under cold stress conditions in Vitis amurensis suggest their involvement in the plant cold stress response. However, the precise molecular mechanisms by which these genes function remain to be elucidated. To fully understand the role of VaKNOX6 and VaKNOX7 in the cold stress response, future studies should focus on functional validation, such as gene knockouts or overexpression experiments, to confirm their roles in cold tolerance. Additionally, exploring the regulatory networks involving these genes, including their interactions with other stress-related genes and hormonal pathways, will provide deeper insights into their molecular mechanisms. Given the importance of cold resistance in crop improvement, particularly in the face of climate change, revealing the functional roles of VaKNOX6 and VaKNOX7 could pave the way for the development of cold-resistant Vitis amurensis varieties.

Conclusions

The KNOX gene family of V. amurensis consists of 10 sequences. An analysis of expression levels revealed that the KNOX genes presented differential expression patterns across various tissues of V. amurensis, suggesting their potential widespread involvement in growth and development. Extreme increases in the expression levels of VaKNOX4 and VaKNOX6 were detected in the seeds. The expression levels of VaKNOX2, VaKNOX3, and VaKNOX5 were extremely elevated in fruits. These observations indicate that VaKNOX2, VaKNOX3, and VaKNOX5 play significant roles in fruit formation and that VaKNOX2, VaKNOX3, and VaKNOX5 can serve as candidate genes for enhancing fruit quality. The expression levels of VaKNOX6 and VaKNOX7 were much higher under cold conditions than under normal conditions. Through in-depth research into the functions of VaKNOX6 and VaKNOX7, we aimed to provide information that could improve the cold resistance of grapevine varieties. In the future, we will elucidate the roles of these genes in key developmental and stress-related pathways through the exploration of gene knockout techniques and CRISPR-based gene editing. Our study provides new insights that could be used in future studies to clarify the function of VaKNOX genes.

Supplemental Information

Supplemental Information 1 Raw data of gene relative expression by qRT-PCR

Supplemental Information 2 MIQE checklist

The authors would like to thank Pengfei Wang for his assistance with the data analysis.

Additional Information and Declarations

Competing Interests

Author Contributions

Data Availability

The authors declare there are no competing interests.

Linling Liu conceived and designed the experiments, performed the experiments, analyzed the data, prepared figures and/or tables, and approved the final draft.

Wenpeng Lu analyzed the data, prepared figures and/or tables, and approved the final draft.

Shutian Fan performed the experiments, authored or reviewed drafts of the article, and approved the final draft.

Yiming Yang conceived and designed the experiments, authored or reviewed drafts of the article, and approved the final draft.

The following information was supplied regarding data availability:

The genomic and proteomic sequence information for V. amurensis is available at the TCMPG 2.0 database: TCMPG20325.

The raw data are available in the Supplemental File.

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
