# Peer review of "Genome-wide identification and characterization of the KNOX gene family in Vitis amurensis"

_PeerJ, doi:10.7717/peerj.19250_

## Round 0.1 · original submission · Major Revisions

Dear Authors

The manuscript cannot be accepted for publication in its current form. It needs substantial revision to meet the journal's standards. The authors are invited to revise the paper, considering all the suggestions made by both reviewers, including reviewer 2 who recommends rejecting the manuscript. Please note that requested changes are required for publication.

With Thanks

**Language Note:** The review process has identified that the English language must be improved. PeerJ can provide language editing services - please contact us at [email protected] for pricing (be sure to provide your manuscript number and title). Alternatively, you should make your own arrangements to improve the language quality and provide details in your response letter. – PeerJ Staff

Reviewer 1 ·

Basic reporting

The manuscript titled "Genome-wide identification and characterization of the KNOX gene family in Vitis amurensis" offers valuable insights; however, substantial revisions are necessary in both the language and technical aspects to improve its clarity and scientific rigor. The specific areas requiring revision are outlined as follows:
In Abstract, while the results mention expression levels and the roles of specific genes, the abstract does not provide sufficient quantitative or comparative context. For example, stating that expression levels are "higher" or "elevated" without a clear baseline or comparison reduces the impact of these findings. Including specific fold-changes or trends would add clarity and rigor to the findings.
The manuscript lacks keywords. Please provide 5-7 distinctive and relevant keywords.
In introduction, please remove unnecessary details related to Vitis amurensis properties rather discuss why is Vitis amurensis species particularly suitable for studying KNOX gene expression or function?
Include a brief explanation of why KNOX gene function differs among species or emphasize the most relevant examples to Vitis amurensis.
Highlight any known or hypothesized connections between KNOX genes and the unique traits of V. amurensis such as its high tannin content or stress tolerance, tying the molecular study more directly to the species’ phenotypic characteristics.
After discussing the functions of KNOX genes in other plants, summarize the gap in knowledge regarding Vitis species and clearly state how the current study aims to fill that gap
The final paragraph introduces the identification and analysis of KNOX genes in V. amurensis but should clearly state why this is significant. Is there a particular gap in knowledge about these genes in grapevines? Will understanding KNOX genes contribute to crop improvement in viticulture?
In materials and methods section, the description of the plant material is not comprehensive enough for reproducibility. For instance, the growth conditions of Vitis amurensis ‘Zuoshan 1’ are not mentioned. Parameters like light, temperature, humidity, and soil type should be described to ensure that the experimental environment is reproducible. Include details of the cultivation conditions (e.g., greenhouse or field conditions, temperature, photoperiod, irrigation regime) as these can affect gene expression and plant physiology.
The protocol for KNOX gene identification is generally sound but lacks some critical details. For example, how were the KNOX1 and KNOX2 models used to filter candidate sequences? Were any threshold parameters set during the BLAST search? How was redundancy handled when merging sequences? Mention specific parameters used in the HMMER and BLAST searches (e.g., E-value thresholds), and clarify how sequence redundancy was removed.
Specify the versions of each software tool used (e.g., HMMER, TBtools).
The method of phylogenetic tree construction is generally accurate but lacks specific details on the neighbor-joining method parameters. Mention if specific substitution models (e.g., JTT, WAG) were applied during the tree-building process in MEGA11. Clarify if "default settings" refer to a specific model, and report how gaps were treated (e.g., pairwise deletion or complete deletion).
The RNA extraction protocol is clearly stated, but it would benefit from mentioning if additional steps were taken to remove contaminants such as DNA, phenol, or polysaccharides, which can be problematic in grapevine tissues.
The qPCR reaction mix components are listed, but it is unclear if no-template controls (NTCs) were included. Specify whether no-template controls (NTCs) and no-reverse transcription controls (NRTs) were used to check for contamination and ensure the validity of the results.
The use of Microsoft Excel and GraphPad Prism for data analysis and visualization is fine, but the statistical tests used to assess RT-qPCR data should be mentioned explicitly (e.g., t-tests, ANOVA). Specify the statistical tests applied to assess the significance of gene expression differences, as this information is critical for understanding how conclusions were drawn.
In Results section, discuss how the presence of these specific cis-elements (e.g., ABRE, G-box, CGTCA) might regulate the expression of KNOX genes in response to environmental cues or developmental stages. For instance, how might the presence of hormone-responsive elements suggest cross-talk between KNOX genes and phytohormones during growth or stress?
The expression analysis reveals differential expression of KNOX genes in tissues and in response to cold, but the biological interpretation is somewhat limited. For instance, no clear hypothesis or conclusion is drawn from the observation that certain genes (e.g., VaKNOX4, VaKNOX6) are highly expressed in buds or seeds, or that others are cold-inducible. Interpret these expression patterns more deeply in terms of known or hypothesized functions of KNOX genes.
Discuss results with key values in terms of percentage increase or decrease (with statistical significance) related to gene expression in various tissues of V. amurensis and expression patterns of the VaKNOX genes in cold condition.
In Discussion section, the manuscript successfully highlights that V. amurensis possesses 10 KNOX genes, a higher number than Arabidopsis and tomato, but fewer than rice and soybean. While this is noteworthy, the analysis of the potential evolutionary significance of this difference could be expanded.
The comparison to other species is appropriate but could benefit from a discussion of functional divergence. How does the presence of a greater or lesser number of KNOX genes impact plant morphology, stress response, or meristematic activity?
The authors state that all KNOX genes are localized to the nucleus, consistent with their role as transcription factors, which is expected. However, more discussion could be added regarding the regulatory networks these KNOX proteins might engage with. Are there known interactors of KNOX proteins in other species that could guide functional hypotheses in V. amurensis?
The lack of domains in certain KNOX proteins (VaKNOX2, VaKNOX7, VaKNOX8) is an interesting observation. However, the functional implications of these domain losses are mentioned only briefly. The ELK domain's role in transcriptional regulation and protein dimerization deserves more exploration, especially concerning how the absence of such domains could alter regulatory interactions or downstream gene expression profiles. Recent studies could be cited to support this reasoning.
The analysis of cis-acting elements in VaKNOX promoters is a strong point of the paper, particularly the identification of light- and stress-responsive elements. However, while it is mentioned that these elements suggest involvement in stress responses, the discussion could be enhanced by incorporating recent findings on how such elements modulate gene expression under specific environmental stresses, such as cold or drought.
The authors make a generalized claim about the role of these cis-elements in abiotic stress responses, but this could be backed by specific references to studies that show how KNOX genes are transcriptionally regulated under stress conditions in other plants.
The tissue-specific expression of VaKNOX genes is discussed adequately, with the observation that Class I genes are highly expressed in buds and tendrils. The linkage to axillary bud regulation and shoot formation is a useful interpretation, but there is a missed opportunity to relate this to broader concepts of plant growth and meristem regulation. For instance, recent research on how KNOX genes interact with hormonal signaling pathways (e.g., auxins, cytokinins) could provide deeper insights into how these genes might control meristem activity in grapevines.
The high expression of VaKNOX genes in fruits and seeds is an intriguing finding. The authors briefly suggest roles in fruit development but should further explore the molecular mechanisms. Is there a known link between KNOX gene expression and fruit ripening, hormone regulation, or seed maturation in other species?
The observation that VaKNOX6 and VaKNOX7 are upregulated under cold conditions is significant and should be discussed in light of recent literature on cold stress regulation in plants. How do KNOX genes potentially modulate cold tolerance mechanisms, such as osmoprotectant synthesis or membrane stabilization? The mention of wheat TaKNOX14-D is a good start, but broader comparisons with other cold-tolerant species could provide a more comprehensive understanding. The discussion could be enriched by mentioning specific pathways or transcriptional networks that KNOX genes might influence during cold stress, such as CBF (C-repeat binding factor) pathways, which are well-established in cold-stress regulation.
The Conclusion summarizes the results well but does not fully leverage the broader significance of the findings. A stronger conclusion would connect the study’s results to potential future applications, such as improving grapevine varieties for cold resistance, enhancing fruit quality through genetic manipulation, or advancing molecular breeding strategies. There is also an opportunity to highlight the need for functional validation of the VaKNOX genes. Future work could explore gene knockouts, overexpression lines, or CRISPR-based gene editing to unravel their roles in key developmental and stress-related pathways.
Specific Comments:
Line 75: "function as a transcriptional regulator" could be rephrased for clarity. Perhaps "KNATM's role as a transcriptional regulator has been well-established through in vivo domain analyses."
After line 79, consider adding a sentence such as, "Given the critical roles of KNOX genes in plant development and their evolutionary conservation, investigating the KNOX gene family in Vitis amurensis could provide insights into the molecular mechanisms driving its unique traits, particularly those relevant to fruit production and stress resilience."
Line 89: "Resveratrol can effectively prevent hyperlipidaemia..." could be rephrased to, e.g., "Resveratrol has been shown to effectively reduce the risk of hyperlipidemia, coronary heart disease, and thrombosis, while also exhibiting anti-cancer properties."
Line 101-102: The phrase "stem samples were collected at minus 20 degrees" is unclear. Specify whether this refers to ambient temperature or if samples were collected under freezing conditions.
Line 118-119: "The protein domains of the KNOX genes... examined using the CDD website and the Pfam databases." Mention specific Pfam IDs used in the analysis and specify whether default search parameters were employed for domain identification.
Line 139-140: The purity and concentration of RNA are measured using a Biochrom Bio Drop Duo. It's important to specify what ratios were considered acceptable for purity assessment.
Line 150: The statement "fluorescence collection at 72°C" is not accurate. Fluorescence is typically collected at the annealing or extension step, which is usually 60°C. Ensure this temperature is correct.
Line 210: "Notably, there were numerous light-responsive elements encompassing 13 distinct types, such as Box 4, G-box, and Sp1, were found." This sentence is redundant and should be rephrased.
Line 219: "By investigating the distinct expression patterns of KNOX genes in various plant tissues, we observed that Class I genes exhibited notably elevated expression levels, particularly in buds and tendrils." Rewrite to improve clarity.

Experimental design

A significant limitation of this study is the lack of clarity regarding the experimental design. The authors did not provide sufficient details about the conditions under which Vitis amurensis was cultivated. Furthermore, the statistical methods used for analyzing gene expression and other results are not adequately described. For robust scientific conclusions, it is essential to mention the statistical tools employed, the sample size, the criteria for significance (e.g., p-values), and the type of statistical tests (e.g., ANOVA, t-tests).

Validity of the findings

The validity of the findings can only be confirmed once the authors provide complete experimental details, including cultivation conditions, and the statistical methods used to analyze the data.

Reviewer 2 ·

Basic reporting

no comment

Experimental design

The authors analysed VaKNOX genes exhibited different expressions under normal and cold conditions by qRT-PCR. But the materials which used in the experiment, were not collected at the same time.

Validity of the findings

There are not many new ideas in the article.

Additional comments

The article only used the grape genome to conduct a relatively basic bioinformatics analysis of the KNOX gene family of grapes. And qRT-PCR was used to analyze the specificity of tissue expression, etc.The innovation and research depth of the article are not high. I don't think it is suitable for publication in this journal.

·

Basic reporting

The manuscript presented by Liu et al., identified and characterized the KNOX gene family in Vitis amurensis. The KNOX gene family is well studied for its role in plant growth and development; thus, the study targets an important gene family for characterization. Although, the study suffers with some issues which are mentioned below:
1. In calculating the fold change, the authors have not calculated the △Ct mean individually for the different samples. Moreover, some data were not even included in mean calculation, for example: in supplementary excel sheet, the △Ct values for cold1, 2 and 3 samples were not included (line 5 to 7). Please rectify this issue.
2. For different developing stages, mean calculation formulae were not applied for △Ct mean; the values are directly pasted in the excel sheet, which makes it difficult to understand and verify the calculation.
3. Please explain, which tissue was used as control for calculating the gene expression in the tissues of developing stages and include this information in the methodology section.
4. Please include the Accession number for Actin gene as it is not provided in the manuscript.
5. According to the analysis of Cis-acting elements in this study, the elements were mostly related to plant hormones, growth, drought and light responses. Then what was the reason for selecting the cold stress for expression analysis in case of these genes? Moreover, authors do not explain the need, and how the results of this experiment are useful, in discussion section. Please answer these points in discussion section.
6. Most of the paragraphs of discussion section do not provide sufficient knowledge and feels like the repetition of result section and the conclusion section suffers with the same issue. I suggest authors to include more relevant information in discussion section. The conclusion needs to be rewritten properly, as there is no need to tell the results again in this section. Only mention the key finding, application aspect and future scope.
7. In discussion section, authors have made some statements which are not supported with suitable citation of literature. For example: authors made a statement “AtSTM is well known for its role in embryonic development, specifically in the establishment of the shoot apical meristem (SAM) and the formation of embryogenic calli. Given this similarity, VaKNOX6 and VaKNOX4 may possess similar biological functions; thus, these proteins could be targeted to increase embryogenic callus formation during the genetic transformation of V. amurensis.”
The cited literature only provides information about role of AtSTM in SAM maintenance and does not discuss the role in the formation of embryogenic calli.
I suggest authors to carefully check the whole discussion for such incidents and cite the suitable literature.
8. Figure legends should explain the figure in detail. please elaborate and enrich the figure legends with more information.

Experimental design

The targeted research question well defined, relevant & meaningful.
Please mention in methodology section that how many biological replicates were used for RT-qPCR analysis? as I can see the data for technical replicate only, in excel sheet.

Validity of the findings

no comment

---

## Round 0.2 · Minor Revisions

Dear Authors
The manuscript still needs a minor revision before publication. The authors are invited to revise the paper considering all the suggestions made by the reviewers. Please note that the requested changes are required for publication.
With Thanks

Reviewer 1 ·

Basic reporting

I appreciate that the authors have addressed most of the queries from the initial review, demonstrating substantial effort in improving the manuscript. However, the revised version still contains some spacing errors, typographical mistakes, and some grammatical errors, and instances of awkward phrasing that need to be corrected. A thorough language review or professional editing is recommended to refine the text.
Additionally, the objectives of the study require further refinement to clearly articulate the research gap and the specific advancements this work brings to the understanding of KNOX genes in Vitis amurensis.
It is also noted that the reference style is inconsistent across the reference list, which should be standardized according to the journal's guidelines.
Lastly, while the authors have improved the clarity of some sections, parts of the discussion remain overly descriptive. Please add a more critical analysis, particularly about the practical applications of VaKNOX genes in stress resistance and grapevine breeding.

Experimental design

The experimental design is appropriate.

Validity of the findings

Findings seems promising.

·

Basic reporting

The authors have incorporated the suggested changes in the manuscript.

Experimental design

no comment

Validity of the findings

no comment

Additional comments

no comment

---

## Round 0.3 · accepted · Accept

Dear Authors,

I am pleased to inform you that the manuscript has improved after the last revision and can be accepted for publication.

Congratulations on accepting your manuscript and thank you for your interest in submitting your work to PeerJ.

With Thanks

Reviewer 1 ·

Basic reporting

The authors have adequately addressed all concerns raised in the previous revision. I recommend the manuscript for acceptance.

Experimental design

Good

Validity of the findings

The findings appear to be scientifically sound.